# ZFHX3 Promotes the Proliferation and Tumor Growth of ER-Positive Breast Cancer Cells Likely by Enhancing Stem-Like Features and *MYC* and *TBX3* Transcription

**DOI:** 10.3390/cancers12113415

**Published:** 2020-11-18

**Authors:** Ge Dong, Gui Ma, Rui Wu, Jinming Liu, Mingcheng Liu, Ang Gao, Xiawei Li, Jun A, Xiaoyu Liu, Zhiqian Zhang, Baotong Zhang, Liya Fu, Jin-Tang Dong

**Affiliations:** 1Department of Genetics and Cell Biology, College of Life Sciences, Nankai University, 94 Weijin Road, Tianjin 300071, China; 1120160366@mail.nankai.edu.cn (G.D.); gma24@wisc.edu (G.M.); 2120171046@mail.nankai.edu.cn (J.L.); 1120170388@mail.nankai.edu.cn (M.L.); agao22@wisc.edu (A.G.); 1120180410@mail.nankai.edu.cn (X.L.); 1120170387@mail.nankai.edu.cn (J.A.); fuchu12@nankai.edu.cn (L.F.); 2Department of Human Cell Biology and Genetics, School of Medicine, Southern University of Science and Technology, 1088 Xueyuan Blvd, Shenzhen 518055, China; wur@sustech.edu.cn (R.W.); liuxy9@sustech.edu.cn (X.L.); zhangzq@sustc.edu.cn (Z.Z.); 3Emory Winship Cancer Institute, Department of Hematology and Medical Oncology, Emory University School of Medicine, 1365-C Clifton Road, Atlanta, GA 30322, USA; baotong.zhang@emory.edu

**Keywords:** ZFHX3, breast cancer, MYC, TBX3, cancer cells stemness

## Abstract

**Simple Summary:**

Breast cancer is a common malignancy, but the understanding of its cellular and molecular mechanisms is limited. The ZFHX3 transcription factor regulates mammary epithelial cells’ proliferation and differentiation by interacting with estrogen and progesterone receptors. Both these receptors play crucial roles in breast cancer development, but whether ZFHX3 also impacts breast cancer is unknown. In this study, the authors aim to determine if ZFHX3 promotes breast cancer cells’ proliferation and tumor growth and explore the underlying cellular and molecular mechanisms. Higher ZFHX3 expression is associated with worse patient survival in breast cancer, ZFHX3 promotes the proliferation and tumor growth of breast cancer cells, and several breast cancer stem cell factors appear to be involved in the role of ZFHX3 in breast cancer growth. The findings suggest that ZFHX3 is a novel oncogenic molecule promoting breast cancer development. Such a molecule could provide novel opportunities for the treatment of breast cancer.

**Abstract:**

Breast cancer is a common malignancy, but the understanding of its cellular and molecular mechanisms is limited. ZFHX3, a transcription factor with many homeodomains and zinc fingers, suppresses prostatic carcinogenesis but promotes tumor growth of liver cancer cells. ZFHX3 regulates mammary epithelial cells’ proliferation and differentiation by interacting with estrogen and progesterone receptors, potent breast cancer regulators. However, whether ZFHX3 plays a role in breast carcinogenesis is unknown. Here, we found that ZFHX3 promoted the proliferation and tumor growth of breast cancer cells in culture and nude mice; and higher expression of *ZFHX3* in human breast cancer specimens was associated with poorer prognosis. The knockdown of *ZFHX3* in ZFHX3-high MCF-7 cells decreased, and ZFHX3 overexpression in ZFHX3-low T-47D cells increased the proportion of breast cancer stem cells (BCSCs) defined by mammosphere formation and the expression of CD44, CD24, and/or aldehyde dehydrogenase 1. Among several transcription factors that have been implicated in BCSCs, MYC and TBX3 were transcriptionally activated by ZFHX3 via promoter binding, as demonstrated by luciferase-reporter and ChIP assays. These findings suggest that ZFHX3 promotes breast cancer cells’ proliferation and tumor growth likely by enhancing BCSC features and upregulating *MYC*, *TBX3*, and others.

## 1. Introduction

Breast cancer is a common malignancy among women [1]. Estrogen receptor alpha (ER) signaling is the defining and driving force in most breast cancers, and selective ER modulator (SERM) therapy is widely used to treat breast cancer. While other signaling pathways also modulate breast carcinogenesis, the expression status of ER, progesterone receptor (PR), human epidermal growth factor receptor 2 (HER2), and the Ki67 proliferation marker stratify breast cancers into several subtypes, such as luminal A (ER^+^ and/or PR^+^/HER2^−^ and low Ki67 index), luminal B (ER^+^ and/or PR^+^ and higher Ki67 index), triple-negative (ER^−^/PR^−^/HER2^−^ with higher Ki67 index), and HER2 enriched [2,3]. Although most breast cancers are luminal tumors and a higher Ki67 index distinguishes luminal B from luminal A tumors [4], ER^+^/PR^−^ luminal tumors have a unique gene expression signature. They are less responsive than ER^+^/PR^+^ tumors to SERM therapy [5]. While PR can be transcriptionally induced by ER and PR can cause or promote breast carcinogenesis [6,7,8], PR also has unique ER-independent functions in luminal breast cancer [7,9]. PR can even be a tumor suppressor in ER-mediated tumor growth of breast cancer [10]. Therefore, it is still an important task to understand the molecular modulators of breast cancer.

ZFHX3 (zinc finger homeobox 3), also known as ATBF1, encodes a large transcription factor with four homeodomains and 23 zinc finger motifs. It has two different transcripts, ATBF1-A and ATBF1-B, with the former having extra residues in the N-terminus [11,12]. ZFHX3 has diverse functions in normal and diseased cells. For example, while ZFHX3 induces neuronal differentiation in both cultured cells and developing mouse brains [11,12,13], it also regulates mammary gland development at different stages via different mechanisms. In pubertal mouse mammary glands, Zfhx3 inhibits cell proliferation and ductal elongation and bifurcation [14]. However, during reproduction, Zfhx3 promotes mammary epithelial cell proliferation, side branching, alveologenesis, and lactogenic differentiation [15,16]. Although higher levels of *ZFHX3-A* mRNA in breast cancer cells are associated with better prognosis, such as smaller tumor size and reduced lymph node metastasis in breast cancer [17], *ZFHX3* is transcriptionally upregulated by both the estrogen/ER and progesterone (Pg)/PR signaling pathways via the binding of ER and PR, respectively, to the *ZFHX3* promoter [15,18,19]. Combining ER and PR’s roles in breast cancer and their interactions with ZFHX3 [14,15,20], it is likely that ZFHX3 plays a role in breast carcinogenesis. However, this hypothesis has not been tested.

In this study, we examined the role of ZFHX3 in breast cancer cells’ proliferation and tumor growth using in vitro and in vivo models. We also explored how ZFHX3 modulates breast cancer growth by focusing on breast cancer stem cell (BCSC) features and ZFHX3′s downstream target genes. We found that ZFHX3 promotes breast cancer cell proliferation and tumor growth, and the underlying cellular and molecular mechanisms involve BCSC-like features and transcriptional activation of *MYC* and *TBX3*. These findings suggest a regulatory role of ZFHX3 in breast carcinogenesis and provide a potential therapeutic opportunity for breast cancer therapy.

## 2. Results

### 2.1. ZFHX3 Enhances Breast Cancer Cell Proliferation and Tumorigenicity

To test whether ZFHX3 modulates breast cancer cell proliferation and tumorigenicity, we first surveyed the levels of endogenous ZFHX3 expression in 5 breast cell lines by western blotting. Among them, MCF-7 had higher expression of ZFHX3, while T-47D, BT-474, and MDA-MB-231 expressed ZFHX3 at lower levels (Appendix A), which is consistent with a previous study [20]. In MCF-7 cells, *ZFHX3* silencing inhibited cell growth (Figure 1a). Consistently, the 3-D colony formation assay showed that both the number and size of spheres were decreased in *ZFHX3* silencing (Figure 1b). In T-47D cells, which express less ZFHX3, ectopic expression of ZFHX3 significantly promoted cell proliferation (Figure 1c) and increased both the size and number of spheres in the Matrigel assay (Figure 1d). ZFHX3 thus plays a promoting role in breast cancer cell proliferation in vitro.

To further test how ZFHX3 impacts breast cancer growth, we injected T-47D cells with ectopic expression of ZFHX3 into the mammary fat pads of nude mice and analyzed their tumorigenicity. ZFHX3 overexpression in T-47D cells significantly increased tumor volume, tumor weight, and the Ki67 proliferation index when compared with control groups (Figure 1e–h). Consistently, *ZFHX3* silencing in MCF-7 cells significantly decreased tumor growth and cell proliferation (Appendix A). Results from the in vivo model further indicate that ZFHX3 increases xenograft tumor growth of breast cancer cells.

We further evaluated the role of ZFHX3 in breast cancer by comparing breast cancers with lower and higher *ZFHX3* mRNA expression levels for patient survival using data from the Cancer Genome Atlas (TCGA) database [21,22]. Patients with lower *ZFHX3* expression in their tumors had longer overall survival time than those with higher *ZFHX3* in their tumors (Figure 1i, *p* = 0.0018). Taking these findings together, we conclude that ZFHX3 promotes breast cancer growth, at least in a subset of patients.

### 2.2. ZFHX3 Is Involved in the Maintenance of BCSC-Like Features

ZFHX3 has been implicated in mammary gland stemness [19], and BCSCs are proposed to drive breast cancer initiation and progression. We thus sought to determine whether ZFHX3 affects BCSC traits. High expression of CD44 and low expression of CD24 (CD44^+^/CD24^−^) are useful for identifying and isolating the cells with stem-like properties in normal mammary tissues and breast carcinomas [23,24,25]. We first compared the expression patterns of ZFHX3 and several genes involved in BCSCs between CD44^+^/CD24^−^ and CD44^−^/CD24^+^ populations of MCF10A cells in the GSE15192 dataset [26]. Interestingly, the CD44^+^/CD24^−^ population expressed higher levels of ZFHX3, along with multiple pluripotency factors such as MYC, TBX3, SOX2, NANOG, and OCT4, than the CD44^−^/CD24^+^ population (Appendix A), suggesting a correlation between ZFHX3 expression and mammary cell stemness.

We then examined the effect of ZFHX3 expression on self-renewal potential, as indicated by the mammosphere forming capability, and the proportions of ALDH^+^ and CD44^+^/CD24^−^ cells. In MCF-7 cells, *ZFHX3* silencing reduced the mammosphere formation efficiency and decreased the population size of ALDH^+^/CD44^+^/CD24^−^ cells (Figure 2a–c). Similarly, silencing *ZFHX3* in MCF-7 cells with two shRNAs, whose knockdown effects were confirmed by western blotting (Appendix A), decreased mammosphere formation (Appendix A) and reduced the ALDH^+^ population when compared to control cells (Appendix A). Conversely, ZFHX3 overexpression was associated with a significant increase in mammosphere forming efficiency and the population size of ALDH^+^/CD44^+^/CD24^−^ cells (Figure 2d–f). These results imply that ZFHX3 is required for maintaining BCSC-like properties of breast cancer cells in vitro.

### 2.3. ZFHX3 Upregulates Stemness Factors Including MYC and TBX3 in Breast Cancer Cells

To further investigate the role of ZFHX3 in the maintenance of BCSC characteristics and explore the underlying mechanisms, we analyzed whether ZFHX3 modulates the expression of multiple pluripotency factors and stemness markers, including MYC, TBX3, OCT4, NANOG, and SOX2, which are involved in the acquisition and maintenance of stem cell properties [27,28,29,30,31,32]. We first detected four of the five stemness factors in xenograft tumors from T-47D cells ectopically expressing ZFHX3 by IHC staining. Overexpression of ZFHX3 significantly increased the ratio of positive cells for each of the four factors (Figure 3a,b). Consistently, western blotting showed that ectopic expression of Flag-tagged ZFHX3 in T-47D cells upregulated MYC, TBX3, OCT4, NANOG, and SOX2 (Figure 3c). In MCF-7 cells, *ZFHX3* silencing by RNAi decreased the expression of MYC and TBX3 as well as that of OCT4, NANOG, and SOX2 (Figure 3d). Thus, the promotion of BCSC properties by ZFHX3 might involve the enhanced expression of stemness factors.

Among these factors, MYC and TBX3 are particularly interesting, because they have been well implicated in BCSC features and breast carcinogenesis [33,34]. TBX3 also promotes mammary epithelial cell stemness and the initiation and progression of breast cancer [28,35]. We, therefore, analyzed the effect of ZFHX3 on the expression of MYC and TBX3. *ZFHX3* silencing in BT-474 and T-47D cells downregulated MYC and TBX3 expression as expected (Appendix A). In MCF-7 cells, estrogen treatment increased ZFHX3 and MYC expression in a dose-dependent manner, while the effect was less evident for TBX3 (Figure 3e). Based on the dose-dependent correlation between ZFHX3 and MYC expression (Figure 3e) and MYC is a well-known estrogen-induced gene, we then examined whether estrogen-induced MYC expression requires ZFHX3. In MCF-7 cells treated with estrogen, which induced dose-dependent upregulation of both ZFHX3 and MYC, knockdown of *ZFHX3* decreased MYC expression compared to the control group (Figure 3f,g). Consistently, ectopic expression of ZFHX3 in T-47D cells further increased estrogen-induced MYC expression compared to the vector control (Figure 3h,i). Estrogen still increased MYC expression in a dose-dependent manner in both MCF-7 cells with ZFHX3 silencing and T-47D cells with ZFHX3 overexpression (Figure 3f–i), suggesting that estrogen signaling and ZFHX3 are two independent events in MYC regulation.

### 2.4. ZFHX3 Transactivates MYC and TBX3 in Breast Cancer Cells

ZFHX3 is a transcription factor, so we investigated whether ZFHX3 directly transactivates *MYC* and *TBX3*. Real-time qPCR showed that *MYC* and *TBX3* mRNAs were decreased by silencing *ZFHX3* in MCF-7 cells (Figure 4a) and were increased by ectopic expression of Flag-tagged ZFHX3 in T-47D cells (Figure 4b).

We then conducted a luciferase promoter-reporter assay to test the effect of ZFHX3 on *MYC* promoter activity. *MYC* promoters of two sizes, one 2 kb, and the other 1 kb upstream to the transcription initiation site, were cloned into the pGL3-Basic luciferase reporter vector for analysis (Figure 4c). *ZFHX3* silencing in MCF-7 cells decreased *MYC* promoter activity for both constructs, particularly the longer one (Figure 4d). Ectopic expression of ZFHX3 in T-47D cells increased *MYC* promoter activity for the longer one but not for the shorter one (Figure 4e), suggesting that ZFHX3 acts on the distal site of the *MYC* promoter.

Indeed, the deletion of fragment C from 1607 to −1437 bp in the distal region of the *MYC* promoter (Figure 4c) prevented ZFHX3 from inducing *MYC* promoter activity in both MCF-7 and T-47D cells (Figure 4d,e). To test whether ZFHX3 binds to the *MYC* promoter, the chromatin immunoprecipitation (ChIP) assay was performed in MCF-7 cells with four sets of primers spanning four fragments (A, B, C, D) of the distal region of *MYC* promoter between −2000 and −1000 bp (Figure 4f). Compared to the control IgG, only fragment C of the promoter from −1607 to −1437 bp was detected in ZFHX3-bound DNA (Figure 4g). Therefore, ZFHX3 directly binds to an element in the 171 bp DNA of the MYC promoter’s distal region to activate *MYC* transcription.

We also analyzed whether ZFHX3 directly activates *TBX3* transcription. Two fragments of the *TBX3* promoter were cloned into the pGL3-Basic luciferase reporter plasmid (Figure 4h, L-TBX3 and S-TBX3), and a luciferase activity assay was performed. RNAi-mediated *ZFHX3* silencing in MCF-7 cells decreased the longer promoter’s activity but not that of the shorter one (Figure 4i), while ZFHX3 overexpressed in T-47D cells increased the activities of both the longer and shorter promoters (Figure 4j). Using the same chromatin precipitates pulled down by the ZFHX3 antibody or its control IgG, PCR with three pairs of primers amplifying three fragments of the *TBX3* promoter region from −1693 to −1074 bp (Figure 4k) demonstrated that fragment b (from −1480 to −1292 bp) was present in ZFHX3-bound DNA, but fragments a and c were not (Figure 4l). These results suggest that ZFHX3 also binds to the *TBX3* promoter to activate its transcription.

### 2.5. Roles of MYC and TBX3 in BCSC Features in the Context of ZFHX3 Expression

We further tested whether the upregulation of MYC and TBX3 by ZFHX3 has functional significance in BCSC properties by analyzing sphere formation and ALDH^+^ cell populations. As expected for established functions of MYC and TBX3 in BCSC regulation, overexpression of MYC or TBX3 alone significantly increased sphere formation efficiency in both MCF-7 and T-47D cells and induced the ALDH^+^ population in T-47D cells. In T-47D cells overexpressing ZFHX3, both MYC and TBX3 were upregulated (Figure 3c), sphere formation was increased, and *MYC* silencing decreased sphere formation in both vector control and ZFHX3 overexpressing cells (Figure 5a,b). In MCF-7 cells with *ZFHX3* silencing, which downregulated both MYC and TBX3 (Figure 3d) and reduced sphere formation (Figure 2a), ectopic expression of MYC failed to increase sphere formation, although it still significantly increased sphere formation in the control group (Figure 5c,d), suggesting that MYC restoration alone is not enough to overcome the effect of *ZFHX3* silencing on spheres.

Sphere formation was also analyzed for TBX3, and similar results were obtained. In T-47D cells with ectopic expression of ZFHX3, knockdown of *TBX3* dramatically reduced spheres in both the control and ZFHX3 groups, and the effect of ZFHX3 expression was not detectable after *TBX3* knockdown (Figure 5e,f). In MCF-7 cells with *ZFHX3* silencing, TBX3 expression significantly increased spheres in the siRNA control group, and the effect was not statistically significant in the *ZFHX3* silencing group as well (Figure 5g,h), suggesting that TBX3 alone is also limited in rescuing the effect of *ZFHX3* silencing on sphere formation.

The ALDH^+^ cell population was also analyzed. In MCF-7 cells with *ZFHX3* silencing, which significantly reduced the percentage of ALDH^+^ cells, expression of TBX3 increased the ALDH^+^ population, but the increase did not completely rescue the effect of *ZFHX3* silencing (Figure 5i). The expression of MYC had even a weaker effect than did TBX3, even though the expression of either MYC or TBX3 significantly increased the ALDH^+^ population in the siRNA control group (Figure 5i). In T-47D cells with ectopic expression of ZFHX3, the ALDH^+^ population was significantly increased. Silencing either *MYC* or *TBX3* by RNAi decreased the ALDH^+^ population in both the vector control and ZFHX3 groups, but neither eliminated the effect of ZFHX3 (Figure 5j). Therefore, it appears that both MYC and TBX3 are involved in ZFHX3 maintained BCSC properties.

## 3. Discussion

In this study, we examined the role of ZFHX3 in the regulation of breast cancer growth and potential underlying mechanisms. We report that ZFHX3 enhances cell proliferation and tumor growth of ER^+^ breast cancer cells. Such a pro-proliferative function involves an enhanced breast cancer stemness and transcriptional upregulation of two factors that have a well-established role in breast cancer, *MYC* and *TBX3*.

ZFHX3 has been suspected of playing a role in breast carcinogenesis because it interacts with ER and PR to regulate gene expression, cell proliferation, cell differentiation, and mammary gland development [15,20]. Both ER and PR have been well implicated in breast cancer. However, direct evidence for such a role of ZFHX3 has been lacking. Our findings in this study indicate that ZFHX3 plays a promoting role in breast cancer, as its knockdown attenuated, while its ectopic expression promoted, the proliferation and tumorigenicity of breast cancer cells in both in vitro and in vivo models (Figure 1 and Appendix A). ZFHX3 also enhanced breast cancer stemness, as indicated by ZFHX3-induced increases in mammosphere formation and the populations of ALDH^+^ or CD44^+^/CD24^−^ cells (Figure 2), and cancer stemness is crucial for tumor initiation and progression. A promoting effect of ZFHX3 on breast cancer is further supported by the correlation of higher *ZFHX3* expression levels with worse patient survival in breast cancer patients of the TCGA database (Figure 1i).

We noticed that ZFHX3 is a tumor suppressor in prostate cancer, as it is frequently mutated in advanced diseases, and its deletion induces or promotes prostatic carcinogenesis in mice [36,37,38]. Even in breast cancer, a tumor suppressor activity has been suggested, as higher levels of the longer transcript of *ZFHX3* (*ZFHX3-A*) correlate with markers of better prognosis such as negative lymph node involvement, lower tumor grade, and smaller tumor size [17]; and in ER^+^/PR^+^ breast cancer cell lines MCF-7 and T-47D cultured in a hormone-free medium, ZFHX3 was inhibitory to estrogen-mediated cell proliferation in 2D culture [20]. These findings appear to contradict the observed tumor-promoting effect of ZFHX3 in this study.

The reasons for such inconsistencies are unknown but are likely very complicated. For example, the two splicing variants of *ZFHX3* (*ZFHX3-A* and *ZFHX3-B*) could have different functions under different contexts, even though we used ZFHX3-A in the ectopic expression experiments. The correlation between *ZFHX3* mRNA expression and the better prognosis was restricted to the *ZFHX3-A* transcript [17]. However, our analysis of TCGA data, where the splicing variants of *ZFHX3* cannot be distinguished, showed that higher levels of *ZFHX3* mRNA expression correlated with worse patient survival (Figure 1i). In the BreastMark database, higher levels of *ZFHX3* mRNA is also associated with poor survival in breast cancer patients [39]. The notion of the two splicing forms having different functions is speculative and remains to be tested.

Interactions of ZFHX3 with other proteins could also modulate protein SUMOylation [40] and regulate multiple pathophysiological processes such as development [15,16,41] and tumorigenesis [37,38]. Such interactions could also make ZFHX3 function in a context-dependent manner. For example, ZFHX3 promotes liver cancer cells’ tumor growth by interacting with HIF1A to enhance angiogenesis [42]. More directly, SUMOylation of ZFHX3 at lysine 2806 is essential for its tumor-promoting effect on the tumorigenicity of MDA-MB-231 cells, as the mutation of lysine 2806 in ZFHX3 prevented MDA-MB-231 cells from forming tumors in nude mice [43].

A more plausible explanation is that ZFHX3 is anti-proliferative when only the estrogen signaling is activated but becomes pro-proliferative when both the estrogen and progesterone signaling is activated. This notion originated from our previous studies of Zfhx3 in mouse mammary gland development at different stages. Zfhx3 modulates multiple hormonal signaling pathways, including estrogen, progesterone, and prolactin [15,16,20,37]. In pubertal mammary glands, estrogen is the primary hormone driving gland development. In such glands, the deletion of Zfhx3 enhances ductal elongation and bifurcation and promotes the proliferation of ER-positive cells [14]. It thus appears that Zfhx3 inhibits the proliferation of mammary epithelial cells when only the estrogen signaling is activated. However, in mature mammary glands, both the estrogen and progesterone signaling is activated, and in such glands, the deletion of Zfhx3 suppresses side-branching, alveologenesis and cell proliferation [15]. It is noteworthy that only when both estrogen and progesterone are present does Zfhx3 promote cell proliferation, as seen in mice with ovariectomy and supplement of estrogen and/or progesterone [15]. Therefore, Zfhx3 exerts contradicting roles in the proliferation of normal mammary epithelial cells, being pro-proliferative when both estrogen and progesterone are present but anti-proliferative when estrogen is dominant.

In our previous study, where ZFHX3 inhibited the proliferation of breast cancer cells [20], cells were cultured in the hormone-free medium for three days to eliminate the effect of other hormones. After that, only the estrogen was added to the medium for analysis [20]. Such a scenario is more analogous to pubertal mammary glands (only the estrogen is dominant), and ZFHX3 being anti-proliferative could thus be expected. In our current study, cells were cultured in a complete medium containing multiple growth factors and hormones and are thus more analogous to adult mammary glands (both estrogen and progesterone are present). ZFHX3′s pro-proliferative function in such a scenario could be expected. Therefore, the role of ZFHX3 in breast cancer appears to be context-dependent. How ZFHX3 executes such a context-dependent function is an important but unanswered question.

Enhancing stemness via transcriptional activation of multiple stem cell factors is a possible mechanism for ZFHX3 to promote breast cancer cells’ proliferation and tumor growth. Flow cytometry analysis demonstrated that ZFHX3 was involved in maintaining the populations of CD44^+^/CD24^−^ and ALDH^+^ cells, both highly enriched for BCSCs and thus have been used to identify BCSCs [44,45,46,47]. *ZFHX3* silencing decreased, while ectopic expression of ZFHX3 increased, both cell populations (Figure 2) and mammosphere formation (Figure 2). In mammary epithelial cells, ZFHX3 appeared to regulate the expansion of progenitor cells [19] and deletion of Zfhx3 in mouse mammary epithelial cells attenuates side-branching and alveologenesis, which involve active proliferation of mammary stem cells [15]. Therefore, although enhanced breast cancer stemness could be a cellular mechanism that is partially responsible for the tumor-promoting effect of ZFHX3 in breast cancer, a firm role of ZFHX3 in the maintenance of BCSCs remains to be established.

Molecularly, MYC and TBX3 could be partially responsible for ZFHX3′s promoting cell proliferation and tumor growth effects. Among the five stem cell markers tested, MYC and TBX3 were more highly induced by ZFHX3 (Figure 3). Indeed, ZFHX3 bound to their promoters to activate their transcription (Figure 4), and re-analysis of the ChIP-seq data from previous studies [16,48,49] using the ChipEnrich program also supported a direct binding of ZFHX3 to the *MYC* promoter. Direct binding of ZFHX3 to *MYC* promoter has also been detected in prostate cancer cells [50]. Both MYC and TBX3 play important roles in breast cancer and BCSCs. MYC is a potent oncogene that alone can reactivate the embryonic stem cell-like program in normal and cancer cells [29,33,51,52,53,54,55] that is overexpressed by various mechanisms in more than half of breast cancers [33,56,57,58,59]; and functionally, MYC indeed contributes to tumor initiation and progression in breast cancer [33,34,60]. At least for MYC, its expression levels correlate with the extents of its effect on cell proliferation. TBX3 is a T-box transcription factor that increases FGF secretion and Wnt signaling activity in mammary glands [61] to regulate self-renewal and differentiation of stem cells [35,62]. TBX3 modulates early embryo and mammary gland development [63,64] and breast cancer initiation and progression via multiple mechanisms such as the paracrine FGF/FGFR/TBX3 signaling pathway [28,35,65,66,67,68]. Breast tumors with higher TBX3 expression levels have greater recurrence rates [64,69]. We should mention that OCT4, NANOG, and SOX2 could also explain the effect of ZFHX3 on BCSC features as they are known stem cell factors, and their expression was also upregulated by ZFHX3 in breast cancer cells (Figure 3).

Although MYC and TBX3 could partially mediate ZFHX3′s promoting effects on cell proliferation and tumor growth, direct evidence is still lacking at this time because findings from the rescue experiments were not conclusive (Figure 5). In T-47D cells where *MYC* or *TBX3* silencing dramatically reduced sphere colony formation, ectopic expression of ZFHX3 showed a trend of increasing sphere formation, but the increase was not statistically significant (Figure 5a,b,e,f). On the other hand, in MCF-7 cells where *ZFHX3* silencing significantly decreased sphere formation and MYC and TBX3 downregulation, re-expression of MYC or TBX3 showed a trend of increasing sphere formation. Still, again the increase was not statistically significant. Multiple reasons could be responsible for these inclusive findings. For example, other molecules could mediate ZFHX3′s function. In T-47D cells, ZFHX3 is likely only one of several factors that upregulate MYC and TBX3; RNAi-mediated silencing of MYC or TBX3 could be too potent.

## 4. Materials and Methods

### 4.1. Cell Lines and Cell Culture

Human breast cancer cell lines MCF-7, T-47D, BT-474, and MDA-MB-231 cells were obtained from ATCC (Manassas, VA, USA). MCF-7 cells were cultured in Dulbecco’s modified Eagle’s medium (DMEM) (Gibco, Grand Island, NY, USA) supplemented with 10% fetal bovine serum (FBS; Gibco). T-47D, BT-474, and MDA-MB-231 cells were cultured in RPMI-1640 medium (Gibco) supplemented with 10% FBS. Cells were cultured at 37 °C with 5% CO_2_. The immortalized non-tumorigenic human breast epithelial cell line MCF10A was purchased from ATCC and cultured in DMEM/F12 medium supplemented with 5% horse serum, 20 ng/mL epidermal growth factor (EGF), 10 µg/mL insulin, 0.5 µg/mL hydrocortisone, and 100 ng/mL cholera toxin.

In experiments involving treatment of cells with estrogen, which was purchased from Sigma-Aldrich (St. Louis, MO, USA), ER^+^ breast cancer cells were cultured in phenol red-free medium supplemented with 5% charcoal dextran-stripped FBS for 24 h, and estrogen was added.

### 4.2. RNA Extraction and Quantitative Real-Time RT-PCR

Total RNA was extracted using the TRIzol reagent (Invitrogen, Carlsbad, CA, USA), and cDNA was synthesized with the MMLV-Reverse Transcriptase system (Promega, Madison, WI, USA). Real-time PCR was performed with the Mastercycler ep realplex system (Eppendorf, Hamburg, Germany) using the SYBR premix Ex Taq (TaKaRa, Tokyo, Japan). RNA expression levels were normalized to glyceraldehyde-3-phosphate dehydrogenase (GAPDH). PCR primers used are shown in Appendix A.

### 4.3. RNA Interference

Cells were seeded onto 6-well plates, grown to 60–80% confluency, and then transfected with small interfering RNAs (siRNAs) using the Lipofectamine RNAiMAX (Invitrogen) according to the manufacturer’s instructions. The siRNA against *ZFHX3*, which targets both *ZFHX3-A* and *ZFHX3-B*, was from a previous study [20]. Sequences for siRNAs are: GGAACUAUGACCUCGACUATT (si*MYC-1*); GAACACACAACGUCUUGGATT (si*MYC-2*); CCUGGAGGCUAAAGAACUUTT (si*TBX3-1*); and GCCUCCACUGUAGGGACAUTT (si*TBX3-2*).

### 4.4. Western Blotting

Cells were washed with PBS and lysed using SDS Lysis buffer (P1016, Solarbio, Beijing, China). Cell lysates containing equal amounts of proteins were separated by 4% (for ZFHX3) or 10% (for all other proteins) SDS-PAGE, then transferred to polyvinylidene fluoride (PVDF) membranes (Millipore, Billerica, MA, USA). The membranes were blocked with 5% nonfat milk at room temperature for 2 h and probed with primary antibodies at 4 °C overnight, then incubated with secondary antibodies for 2 h at room temperature. They were then visualized using WesternBright ECL (Advansta, Menlo Park, CA, USA), and protein signals were detected with the luminescent Image Analyzer (Jun Yi Dong Fang, Beijing, China). Protein band intensities were determined using the ImageJ program and ratios of protein band intensities to those of their loading controls were calculated as previously described [70]. ZFHX3, whose molecular weight is 404 KDa, and the antibody for ZFHX3 has been described in our previous study [20]. Other antibodies included β-actin (1:8000, 3700, Cell Signaling, Danvers, MA, USA); MYC (1:1000, 9402, Cell Signaling); TBX3 (1:1000, ab154828, Abcam, Cambridge, MA, USA); OCT4 (1:1000, 2750, Cell Signaling); NANOG (1:1000, 4903, Cell Signaling); SOX2 (1:1000, ab92494, Abcam); ERα (1:1000, sc-53493, Santa Cruz, CA, USA); FLAG (1:3000, SAB4200071, Sigma); and Myc-Tag (1:500, 06-549, Sigma). Original unedited blots can be found at Appendix A–S12.

### 4.5. Establishment of Cell Lines

MISSION^TM^ shRNAs targeting *ZFHX3* in pLKO.1-puro plasmid and vector control plasmid were purchased from Sigma. Lentiviral particles were produced in 293T cells by cotransfecting pLKO.1 with pMD2.G and psPAX2 plasmids using the FuGENE 6 transfection reagent according to the manufacture’s protocol (Promega). MCF-7 cells were seeded in 6-well plates and cultured with fresh medium containing 8 µg/mL of Polybrene (Sigma), and 1 mL of lentiviral solution was added dropwise. After 6 h of viral infection, the virus-containing medium was replaced with a fresh medium containing puromycin at 2 µg/mL (Sigma) for 3 days to select cells stably expressing shRNAs. MCF-7 cells expressing shZFHX3 were maintained in a medium containing 1 µg/mL puromycin. Sequences of shRNAs against *ZFHX3* are CCGGGCCAGGAAGAATTATGAGAATCTCGAGATTCTCATAATTCTTCCTGGCTTTTT (sh*ZFHX3-1*) and CCGGCCCTTTAGTTTCCACAGCTAACTCGAGTTAGCTGTGGAAACTAAAGGGTTTTT (sh*ZFHX3-2*), which have been previously described [16].

T-47D cells were transfected with pcDNA3.0-Flag, and Flag-tagged ZFHX3 using the Lipofectamine 2000 Transfection Reagent (Invitrogen), and transfected cells were selected with 1 mg/mL G418 for 14 days. ZFHX3 expression in transfected T-47D cells was confirmed by western blotting.

### 4.6. Cell Growth and Proliferation Assay

Cell growth and proliferation were analyzed by the sulforhodamine B (SRB) staining assay. MCF-7 cells were seeded onto 24-well plates and transfected with siRNAs against *ZFHX3* or control. T-47D cells were seeded on 24-well plates and transfected with pcDNA3.0-Flag or Flag-tagged ZFHX3. The culture medium was replaced every other day. At desired time points, cells were fixed by 10% trichloroacetic acid (TCA) for 1 h at 4 °C, stained with 0.4% SRB (Sigma), and washed with 1% acetic acid. Stained cells were dissolved, and the absorbance was measured.

### 4.7. Sphere Formation Assay in Matrigel (3D Culture)

The growth factor reduced Matrigel (354230, Corning, NY, USA) was used for this assay. Briefly, cells were seeded onto 6-well plates and transfected with siRNA or plasmids. Eight-well chamber slides (PEZGS0896, Millipore) were loaded with 40 µL Matrigel, solidified at 37 °C for at least 30 min. A total of 2500 (Figure 1) or 1000 (Figure 5) treated cells in the mixture of medium containing 10% FBS and 2% Matrigel were overlaid on the gel and were then grown for 2 weeks, with the medium renewed every 2 days. Images of spheres were subjected to the ImageJ computer program [70] to determine the spheres’ diameter and number.

### 4.8. Mammosphere Formation Assay

A single-cell suspension was obtained and plated in ultra-low attachment plates (3471, Corning) at a density of 1000 cells per well. Cells were then cultured in a complete mammosphere culture medium (05620, Stem Cell Technology, Vancouver, BC, Canada) for 10–14 days. The number of mammospheres with a diameter >60 µm was determined using the Image J program.

### 4.9. Flow Cytometry Assay

A single-cell suspension was washed with phosphate-buffered saline (PBS), resuspended in PBS (10^6^ cells per 100 µL), and incubated with the antibodies of human cell surface markers PE-CD24 (560991, BD Pharmingen, San Diego, CA, USA) and APC-CD44 (559942, BD Pharmingen) for 1 h on ice in the dark. Control samples were incubated with isotype-matched control IgG, PE-anti-CD24 (555574, BD Pharmingen), APC-anti-CD44 (555745, BD Pharmingen) for 1 h on ice in the dark. Unbound antibodies were washed off, and labeled cells were analyzed using the BD FACS-Calibur (BD Biosciences, San Jose, CA, USA).

### 4.10. Aldefluor Assay

The Aldefluor assay was performed using the Aldefluor kit from Stem Cell Technology following the manufacturer’s protocol (01700, Stem Cell Technology). Briefly, 10^6^ cells were resuspended in 1 mL Aldefluor buffer, and 5 µL of ALDH substrate (BAAA) was added and mixed. 500 µL of the cell suspension was immediately transferred to another tube containing 5 µL of diethylaminobenzaldehyde (DEAB), an ALDH inhibitor, mixed evenly, and used as the negative control. Cells in both tubes were incubated for 40 min at 37 °C, washed, resuspended in 500 µL Aldefluor buffer, and then analyzed with the BD FACS-Calibur (BD Biosciences).

### 4.11. Promoter-Luciferase Assay

Cells were seeded in 24-well plates, cultured overnight, and then transfected with the promoter-reporter plasmids using the Lipofectamine 2000 Transfection Reagent (Invitrogen). After 48 h, cells were collected in reporter lysis buffer (E3971, Promega), and the luciferase activity was measured using the dual-luciferase assay kit (Promega) with a luminometer ((Tristar LB941, Berthold Technologies, BadWild, Germany). Primer sequences for cloning the promoters of *MYC* and *TBX3* are listed in Appendix A.

### 4.12. Chromatin Immunoprecipitation (ChIP) Assay

ChIP assays were performed by using the Simple ChIP Enzymatic Chromatin Immunoprecipitation Kit (9003, Cell Signaling Technology) according to the manufacturer’s instructions. ChIP products were detected by regular PCR. Primer sequences for *MYC* and *TBX3* promoters are shown in Appendix A.

### 4.13. Tumorigenesis Assay

T-47D cells stably transfected with Flag-tagged vector or Flag-ZFHX3 plasmid were resuspended in PBS/Matrigel (354234, Corning) at 1: 1 ratio and injected into the inguinal mammary gland of BALB/c nu/nu athymic mice (4–6-week-old female) (*n* = 6) at 5 × 10^6^ cells per gland, with the vector control cells injected into the right side mammary fat pad and Flag-ZFHX3 cells into the left mammary fat pad. Estrogen pellets were implanted in the back of the neck. For Supplementary Figure 2, 1 × 10^7^ MCF-7 cells of shZFHX3 or control (shCon) were suspended in 100 µL of PBS/Matrigel (354234, Corning) mix (1:1) and injected subcutaneously into mammary fat pads of female nude mice. Estrogen pellets were implanted into the back of the neck. The length (L) and width (W) of xenograft tumors were measured once a week. At the end of the experiment, mice were euthanized; and tumors were isolated, fixed in 4% paraformaldehyde, and subjected to IHC or hematoxylin and eosin (HE) staining.

### 4.14. Immunohistochemical (IHC) Staining

Fixed tumors were paraffin-embedded and sectioned, and sections were deparaffinized, rehydrated following standard procedure, and IHC staining was performed with an IHC kit from MXB (KT-5001, Fuzhou, Fujian, China). Tissue sections were then stained with hematoxylin, dehydrated, and mounted. In T-47D cells with ZFHX3 overexpressing, ZFHX3 was localized in both the cytoplasm and nucleus (Figure 1h), whereas Ki67, MYC, TBX3, OCT4, and SOX2 were localized in the nucleus (Figure 3a,b). For the quantification of ZFHX3 expression, slides were scanned using an Aperio VERSA 8 Scanner System (Leica Microsystems, Wetzlar, Germany). The ImageJ program was then used to determine the intensity of positively stained signal (brown) and the total nuclei (blue) in the tissue area. The average intensity per nucleus (cell) was then calculated. For the other five proteins, images of stained tissue sections were taken at 20× magnification using either the same Aperio VERSA 8 system or an Aperio ScanScope XT system (Leica). The numbers of positively stained cells and total cells were then determined from 3–5 images using the ImageJ program. The ratio of positively stained cells to total cells was used for comparison.

The antibodies used in IHC analysis included ZFHX3 (1:1000, PD010, MBL, Nagoya Aichi, Japan), Ki67 (1:2000, ab15580, Abcam), MYC (1:1000, ab32072, Abcam), TBX3 (1:100, SRP08533, Tianjin Saierbio, Tianjin, China), OCT4 (1:1000, 2750, Cell Signaling), and SOX2 (1:1000, ab92494, Abcam).

### 4.15. Bioinformatic and Survival Analyses

Gene expression profile of GSE15192 was downloaded from the Gene Expression Omnibus (GEO) [71,72] and applied for heat-map via R language software (Windows v3.6.3, https://www.r-project.org/). Correlation of survival with different expression levels of *ZFHX3* in patients with breast cancer was analyzed and visualized by the embedded UALCAN tool in the TCGA database [21,22].

### 4.16. Statistical Analysis

All in vitro experiments were repeated at least twice. Statistical analyses were performed using the SPSS statistical software (SPSS, Version 20; IBM, Armonk, NY, USA). Two-tailed Student’s *t*-test was used to compare two groups, and one-way ANOVA was used for comparison of more than two groups. All quantitative data are expressed as mean ± SD. *p* values less than 0.05 were considered statistically significant.

## 5. Conclusions

In summary, these findings suggest that transcription factor ZFHX3 plays a promoting function in breast cancer cells’ proliferation and tumor growth involving the BCSC properties. Molecularly, ZFHX3 could bind to the promoters of *MYC* and *TBX3* to activate their transcription, which could then partially mediate ZFHX3′s effects on cell proliferation and tumor growth. These findings thus suggest a novel molecular mechanism underlying breast carcinogenesis that could help develop therapeutic strategies for the treatment of breast cancer.

## Figures and Tables

**Figure 1 cancers-12-03415-f001:**
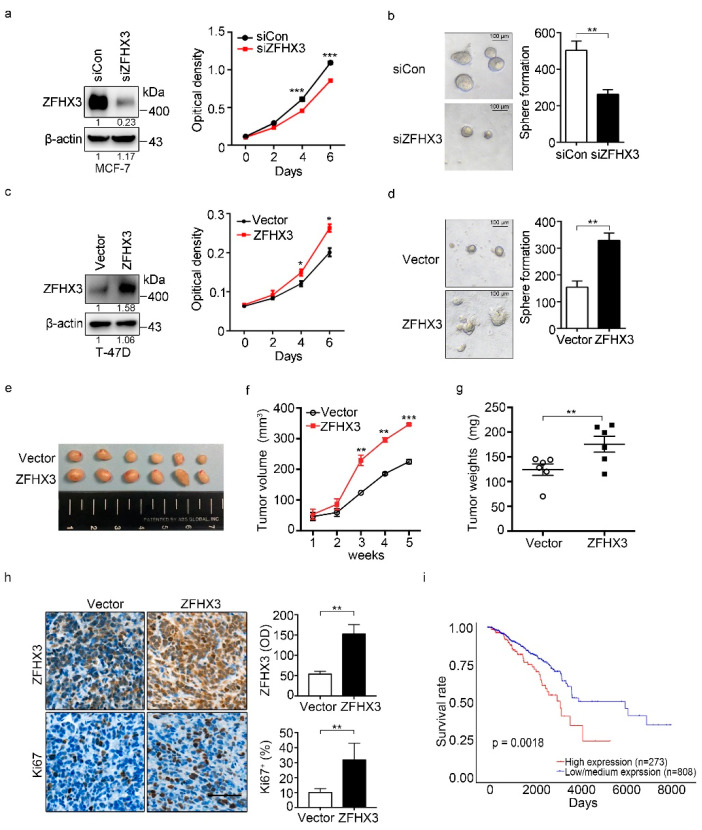
ZFHX3 enhances colony formation and tumorigenicity of ER^+^ breast cancer cells. (**a**,**b**) *ZFHX3* silencing by RNAi inhibited colony formation in 2D culture (**a**), as determined by the sulforhodamine B (SRB) assay, and sphere formation in Matrigel (**b**), as indicated by representative images of spheres (left) and the numbers of spheres with a diameter >75 µm (right) in MCF-7 cells. siCon, control siRNA; siZFHX3, siRNA against *ZFHX3*. (**c**,**d**) Ectopic expression of ZFHX3 promoted colony formation in the 2D SRB assay (**c**) and 3D sphere formation in Matrigel (**d**) in T-47D cells. (**e**–**g**) ZFHX3 expression in T-47D cells also promoted their subcutaneous tumor formation in nude mice, as indicated by tumor images (**e**), tumor growth curve (**f**), and tumor weight (**g**). (**h**) Enhanced tumor growth by ZFHX3 was accompanied by increased cell proliferation, as indicated by representative images of IHC staining of ZFHX3, Ki67 in xenograft tumors, and the quantification of ZFHX3-positive cells (up), Ki67-positive cells (down). Scale bar, 50 µm. (**i**) Higher mRNA levels of *ZFHX3* are associated with worse survival in patients with breast cancer in the TCGA database, as analyzed by survival analysis. * *p* < 0.05; ** *p* < 0.01; *** *p* < 0.001. Ratios of protein band intensities to those of their loading controls, with the control sample’s normalized to 1, are shown under bands in western blots (**a**,**c**). Uncropped western blot images are available in Appendix A.

**Figure 2 cancers-12-03415-f002:**
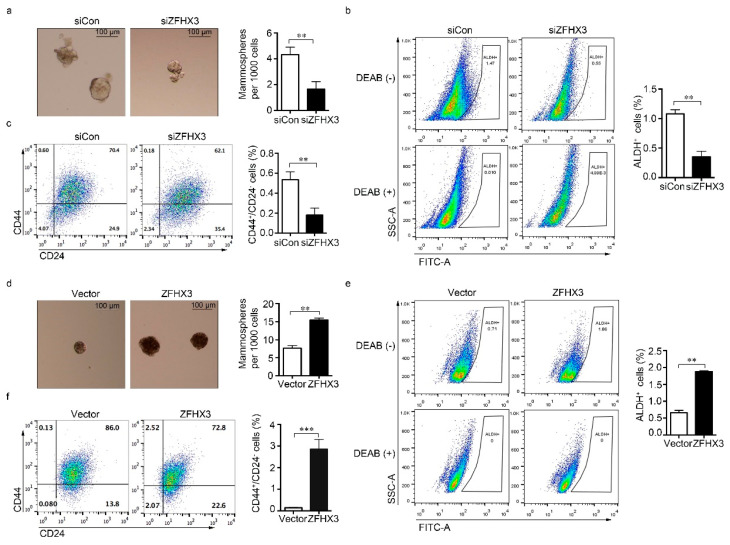
ZFHX3 maintains stem-like characteristics of ER^+^ MCF-7 and T-47D breast cancer cells. (**a**–**c**) RNAi-mediated knockdown of *ZFHX3* in MCF-7 cells decreased mammosphere forming ability (**a**), as indicated by images of mammospheres (left) and their quantification (right) from the mammosphere formation assay, and the populations of ALDH^+^ (**b**) and CD44^+^/CD24^−^ (**c**) cells, as analyzed by flow cytometry. (**d**–**f**) Ectopic expression of ZFHX3 in T-47D cells increased mammosphere forming ability (**d**) and the populations of ALDH^+^ (**e**) and CD44^+^/CD24^−^ (**f**) cells, as analyzed by flow cytometry. siCon, control siRNA; siZFHX3, siRNA against *ZFHX3*. ** *p* < 0.01; *** *p* < 0.001. The blank and isotype controls are shown in Appendix A.

**Figure 3 cancers-12-03415-f003:**
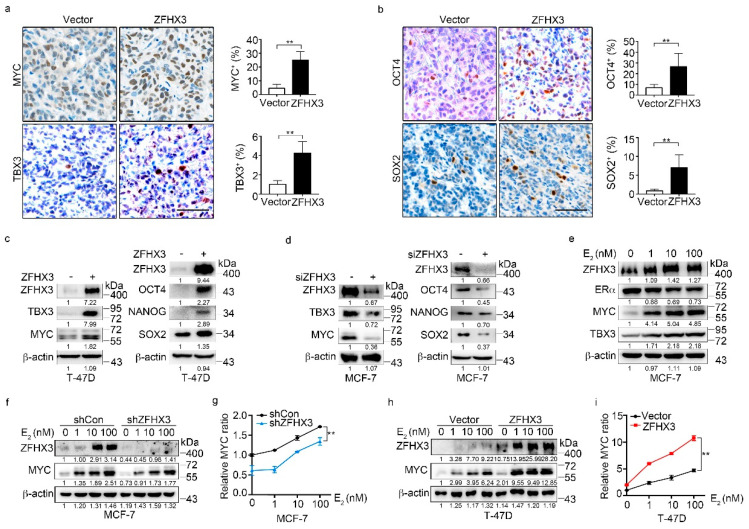
ZFHX3 induces the expression of MYC and TBX3 in ER^+^ MCF-7 and T-47D breast cancer cells. (**a**,**b**) Detection of MYC, TBX3, OCT4, and SOX2 by IHC staining in xenograft tumors of T-47D cells ectopically expressing ZFHX3, as shown by representative IHC images (left) and quantification of cells with positive staining (right). Scale bar, 50 µm. (**c**,**d**) Ectopic expression of ZFHX3 in T-47D cells increased (**c**), while *ZFHX3* silencing in MCF-7 cells decreased (**d**), the expression of MYC, TBX3, OCT4, NANOG, and SOX2, as detected by western blotting analysis. (**e**) Estrogen (E_2_) increased MYC and TBX3 in MCF-7 cells, as detected by western blotting. (**f**–**i**) *ZFHX3* silencing in MCF-7 cells attenuated, while ectopic expression of ZFHX3 in T-47D cells enhanced, E_2_-mediated MYC expression, as detected by western blotting (**f**,**h**) and quantified by relative MYC ratio (**g**,**i**). Cells were treated with E_2_ for 24 h at the indicated concentrations. siCon, control siRNA; siZFHX3, siRNA against *ZFHX3*. ** *p* < 0.01. (**a**,**b**) IHC staining was done in xenograft tumors. (**c**–**i**) cells were cultured in 2D plastic surface and examined with western blotting. Ratios of protein band intensities to those of their loading controls, with the control sample’s normalized to 1, are shown under western blot bands (**c**–**f**,**h**). Uncropped western blot images are available in Appendix A.

**Figure 4 cancers-12-03415-f004:**
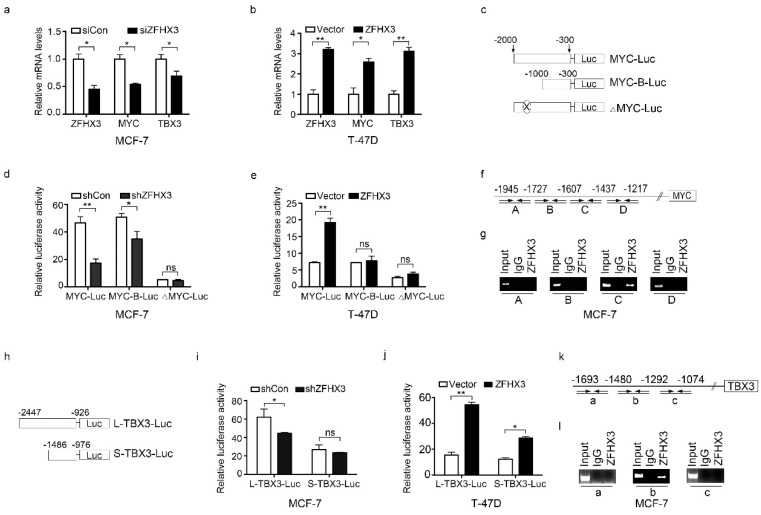
ZFHX3 activates *MYC* and *TBX3* transcription by binding to their promoters in ER^+^ MCF-7 and T-47D breast cancer cells. (**a**,**b**) *ZFHX3* silencing in MCF-7 cells decreased, while ectopic expression of ZFHX3 in T-47D (**b**) cells increased, mRNA levels of *MYC* and *TBX3*, as detected by real-time PCR. (**c**) Schematic for the construction of different *MYC* promoter-luciferase reporter plasmids. MYC-Luc, 2 kb promoter sequences; MYC-B-Luc, 1 kb; ΔMYC-Luc, the 171 bp from −1607 to −1437 bp upstream to the transcription initiation site was deleted. (**d**,**e**) *ZFHX3* silencing in MCF-7 cells (**d**) and ZFHX3 expression in T-47D cells (**e**) had different effects on the activities of different MYC promoter constructs, as detected by luciferase reporter assay. (**f**,**g**) Location of PCR primer pairs, indicated by arrows, that define fragments A, B, C, and D in the *MYC* promoter (**f**) and detection of ZFHX3 binding to the C region (−1607 to −1437 bp) by ChIP-PCR. (**h**) Schematic for the construction of *TBX3* promoter-reporter plasmids. L-TBX3-Luc and S-TBX3-Luc contain 1.52 kb and 0.51 kb DNA sequence, respectively, upstream to the *TBX3* transcription initiation site. (**i**,**j**) Effects of *ZFHX3* silencing in MCF-7 cells (**i**) and ZFHX3 expression in T-47D cells (**j**) on the two TBX3 constructs’ luciferase activities, as detected by luciferase reporter assay. (**k**,**l**) Location of PCR primer pairs, indicated by arrows, that define fragments a, b, and c in the *TBX3* promoter (**k**) and detection of ZFHX3 binding to the b region (−1480 to −1292 bp) by ChIP-PCR (**l**). siCon, control siRNA; siZFHX3, siRNA against *ZFHX3*. ns, not significant; * *p* < 0.05; ** *p* < 0.01. Cells were cultured on a 2D plastic surface, then examined in real-time PCR, promoter-luciferase assay, and ChIP assay.

**Figure 5 cancers-12-03415-f005:**
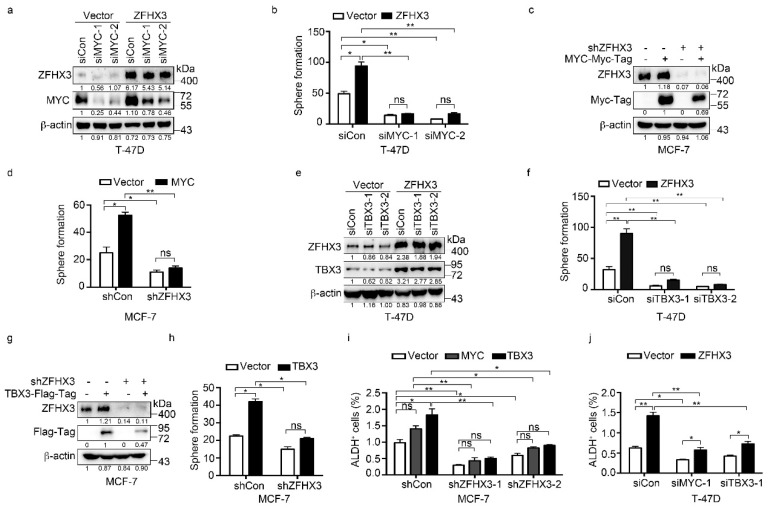
MYC and TBX3 play a role in ZFHX3-mediated sphere formation in Matrigel and ALDH^+^ cell population in ER^+^ MCF-7 and T-47D breast cancer cells. (**a**,**b**) In T-47D cells ectopically expressing ZFHX3, *MYC* silencing, as confirmed by western blotting in 2D culture (**a**), and culture in Matrigel (3D culture) that attenuated sphere formation in both ZFHX3 and vector control groups (**b**). (**c**,**d**) In MCF-7 cells with *ZFHX3* silencing, MYC ectopic expression, as confirmed by western blotting in 2D culture (**c**), and culture in Matrigel (3D culture) that did not significantly increase sphere number (**d**). (**e**,**f**) In T-47D cells ectopically expressing ZFHX3, *TBX3* silencing, as confirmed by western blotting in 2D culture (**e**), and culture in Matrigel (3D culture) that attenuated sphere formation (**f**). (**g**,**h**) In MCF-7 cells with *ZFHX3* silencing, TBX3 ectopic expression, as confirmed by western blotting in 2D culture (**g**), and culture in Matrigel (3D culture) that did not significantly increase sphere number (**h**). (**i**,**j**) ALDH^+^ cells were then detected by flow cytometry in MCF-7 cells with *ZFHX3* silencing and MYC or TBX3 overexpression (**i**) and T-47D cells with ZFHX3 overexpression and *MYC* or *TBX3* knockdown (**j**). ns, not significant; * *p* < 0.05; ** *p* < 0.01. Cells were cultured on a 2D plastic surface. siCon, control siRNA; siZFHX3, siRNA against *ZFHX3*. Ratios of protein band intensities to those of their loading controls, with the control sample’s normalized to 1, are shown under western blot bands (**a**,**c**,**e**,**g**). Uncropped western blot images are available in Appendix A.

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
