# Peer review of "ZFHX3 Promotes the Proliferation and Tumor Growth of ER-Positive Breast Cancer Cells Likely by Enhancing Stem-Like Features and MYC and TBX3 Transcription"

_cancers, 2020, doi:10.3390/cancers12113415_

Round 1

Reviewer 1 Report

The authors have clearly followed the comments of the 3 referees and have now greatly improved the manuscript. Apart from the below, the majority of my comments have been answered (even though while the IHC data being now concordant, the re-analysis gives now different values).

The issue with point 5 is not fully resolved but agree with the author that I will leave the decision to the Editor. My initial comment was dismissed by bringing up the existence of two splicing variants. When I question this, this is now dismissed and a new issue is brought up: the presence of progesterone in the culture media in this study compared to previous one. The authors write several paragraphs commenting on this. However, the Methods clearly show that no progesterone is added to the media. I am at a loss of what to make of this and will the Editor decide on this issue.

Western blots: I would comments that the variations I highlighted in my previous review are beyond blots to blots variations. The authors have not included the blots as they mention in the second paragraph of page 2 of their response: "For band intensities, each blot has its exposure condition in western blotting; and one cannot compare one protein’s band intensities among different gels, not to mention different proteins from different blots. We have now included all original images of western blotting, which helps compare expression levels."

Author Response

Reviewer #1:

Comment 1: The issue with point 5 is not fully resolved but agree with the author that I will leave the decision to the Editor. My initial comment was dismissed by bringing up the existence of two splicing variants. When I question this, this is now dismissed and a new issue is brought up: the presence of progesterone in the culture media in this study compared to previous one. The authors write several paragraphs commenting on this. However, the Methods clearly show that no progesterone is added to the media. I am at a loss of what to make of this and will the Editor decide on this issue.

Response 1: Regarding the comment that "no progesterone is added to the media," it is correct that we did not add progesterone to the medium. We proposed that normal serum contains many kinds of growth factors and hormone analogs, some of which could have an effect similar to progesterone. As the reviewer mentioned, we spent several paragraphs explaining the contradiction between this study and the previous one and would like to wait for the editor to decide.

Comment 2: Western blots: I would comments that the variations I highlighted in my previous review are beyond blots to blots variations. The authors have not included the blots as they mention in the second paragraph of page 2 of their response: "For band intensities, each blot has its exposure condition in western blotting; and one cannot compare one protein's band intensities among different gels, not to mention different proteins from different blots. We have now included all original images of western blotting, which helps compare expression levels."

Response 2: Regarding the comment on our sentence that "We have now included all original images of western blotting, which helps compare expression levels," we meant that all the original images of western blotting had been uploaded to the system following the journal's instructions. Such a request was not there in the previous submission of this manuscript seven months ago.

Reviewer 2 Report

The manuscript of Dong and colleagues suggests that ZFHX3 promotes breast cancer proliferation partially enhanced through MYC and TBX3. The authors acceded and tried to incorporate most of this reviewer’s comments. Nevertheless, during the revision process the main message of the manuscript has to be changed as it was and partially is still overstated, as crucial experiments are missing. Moreover, an important aspect of medium composition and its influence on ZFHX3 surfaced, which actually is very intriguing and deserves further attention and experiments around it included into the manuscript. Additionally, the controversy around the isoforms of ZFHX3 makes the results inconsistent and as the authors admitted worth considering as they currently follow up on that. I would suggest waiting for the results of those experiments and include them into the manuscript as they will be crucial to either confirm the author’s hypothesis or not. Once they are included the manuscript is ready to publish after addressing the minor comments mentioned below as well.

-Introduce Her2 in line 40

-State which Isoforms of ZFHX3 are targeted by the siRNA used in Fig1a and b. That is crucial to the manuscript as that  the authors use as an explanation for the discrepancy to their earlier studies.

-306 missing a full stop.

-Lane 337 ‘How ZFHX3 executes such a context-dependent function is an important but answered question’ authors mean unanswered. Unanswered, as this is only speculation so far and no evidence is presented in the manuscript.

-Please add to the discussion that OCT4, NANOG, and SOX2 could be factors that as well might explain stemness as they are known stem sell factors as well and are often used to induce pluripotency.

-Treatment with tamoxifen or fulvestrant needs to be removed from methods, no experiment are shown with those treatment.

Author Response

Reviewer #2:

Comment 1: The manuscript of Dong and colleagues suggests that ZFHX3 promotes breast cancer proliferation partially enhanced through MYC and TBX3. The authors acceded and tried to incorporate most of this reviewer's comments. Nevertheless, during the revision process the main message of the manuscript has to be changed as it was and partially is still overstated, as crucial experiments are missing.

Response 1: We appreciate the reviewer for recognizing that "the main message of the manuscript has to be changed as it was," which refers to our effort in downplaying the role of stemness and MYC/TBX3 upregulation as mechanisms of ZFHX3 promoting breast cancer. The reviewer went on saying that "partially is still overstated, as crucial experiments are missing." Based on this statement, I could not figure out what he/she meant by saying that "partially is still overstated, as crucial experiments are missing." If he/she meant the experiments suggested in the other comments, my responses are provided below to address those comments.

Nevertheless, we have modified the title, abstract, and some figure legends to further address the issue.

  1. The title has been revised again, from "ZFHX3 promotes breast cancer cell proliferation and tumor growth in part by enhancing stem-like features and MYC and TBX3 transcription" to "ZFHX3 promotes the proliferation and tumor growth of ER-positive breast cancer cells likely by enhancing stem-like features and MYC and TBX3 transcription".
  2. The last sentence in the abstract (Page 1, Line 31-33) has been revised to the following: "These findings suggest that ZFHX3 promotes breast cancer cells' proliferation and tumor growth likely by enhancing BCSC features and upregulating MYC, TBX3, and others".
  3. We have changed "breast cancer cells" in Figure legends to "ER+ MCF-7 and T-47D breast cancer cells".

Comment 2: Moreover, an important aspect of medium composition and its influence on ZFHX3 surfaced, which actually is very intriguing and deserves further attention and experiments around it included into the manuscript.

Response 2: We appreciate the reviewer's recognition of this intriguing point. We realized that we should have discussed the aspect of medium composition in the very beginning version of this manuscript submitted seven months ago. However, as discussed in the previous version, we have already demonstrated in mouse mammary glands that Zfhx3 is pro-proliferative during pubertal development (PLoS One 7:e51283, 2012), during which only the estrogen is active, but is anti-proliferative during adulthood and pregnancy (J. Genet. Genomics 46:119-31, 2019), during which both estrogen and progesterone are active. Therefore, the more important question is how ZFHX3 exerts opposing functions in cell proliferation under different estrogen and progesterone conditions.

I hope the reviewer will agree that addressing how ZFHX3 exerts opposing functions under the two conditions is not a simple task. For example, we hypothesize that ZFHX3 is likely a scaffold protein that provides a platform for other nuclear factors to function. When the nuclear factor is ERβ in prostate cancer cells (Oncogenesis 8:28, 2019) or ERα in breast cancer cells (J. Biol. Chem. 285:32801-9, 2010), ZFHX3 is anti-proliferative. When the nuclear factor is ERα plus PR in breast cancer cells (this manuscript) or HIF1A in liver cancer cells (J. Biol. Chem. 295:7060-74, 2020), it is pro-anti-proliferative. It will take a considerable effort, certainly beyond the current manuscript's scope, to address this hypothesis. For example, it will be necessary to determine how ZFHX3 is associated with chromatin or other components of the nucleus, what proteins ZFHX3 interact with, and how it makes ERα and ERα plus PR function differently, just name a few.

I should also mention that we were given only one week by the editor to revise the manuscript, which is not enough for any experiments.

Comment 3: Additionally, the controversy around the isoforms of ZFHX3 makes the results inconsistent and as the authors admitted worth considering as they currently follow up on that. I would suggest waiting for the results of those experiments and include them into the manuscript as they will be crucial to either confirm the author's hypothesis or not.

Response 3: I should clarify that it is entirely our speculation that the two ZFHX3 isoforms could function differently. We do not have any evidence. In all our published and unpublished studies using ectopic ZFHX3 expression, including those in the current manuscript, only the A isoform was used. In addition, we have been using two antibodies against ZFHX3, one is specific to the A isoform as it was raised using peptide from A-specific sequence, and the other detects both isoforms as it is against a peptide shared by both isoforms. We have not noticed a difference in ZFHX3's band size between the two antibodies on previous western blots. Therefore, although we cannot exclude the possibility of the two isoforms having different functions, we do not have any supporting evidence. The siRNA used in this study was from a previous publication and targets both ZFHX3-A and ZFHX3-B, so the related results cannot help solve this issue.

In this regard, I have revised the last sentence of Paragraph 3 on Page 10 (Line 307-308) to the following: "The notion of the two splicing forms having different functions is speculative and remains to be tested." This revision should help avoid misleading readers to think that there is evidence for a difference in the two isoforms' function.

Minor comments:

Comment 1: Introduce Her2 in line 40.

Response 1: HER2 is now spelled out (the human epidermal growth factor receptor 2) (Page 1, line 40-41).

Comment 2: State which Isoforms of ZFHX3 are targeted by the siRNA used in Fig1a and b. That is crucial to the manuscript as that  the authors use as an explanation for discrepancy to their earlier studies.

Response 2: The siRNA against ZFHX3 in the manuscript targets both the isoforms of ZFHX3, which is now indicated by the following sentence in Methods (Page 12, Line 406-407): "The siRNA against ZFHX3, which targets both ZFHX3-A and ZFHX3-B, was from a previous study [20]".

Comment 3: 306 missing  a full stop.

Response 3: Added.

Comment 4: Lane 337 'How ZFHX3 executes such a context-dependent function is an important but answered question' authors mean unanswered. Unanswered, as this is only a speculation, and no evidence is available at this time.

Response 4: Corrected.

Comment 5: Please add to the discussion that OCT4, NANOG, and SOX2 could be factors that as well might explain stemness as they are known stem sell factors as well and are often used to induce pluripotency.

Response 5: We have added the following sentence to Paragraph 2 on Page 11 (Line 367-370): "We should mention that OCT4, NANOG, and SOX2 could also explain the effect of ZFHX3 on BCSC features as they are known stem cell factors, and ZFHX3 also upregulated their expression in breast cancer cells (Figure 3)".

Comment 6: Treatment with tamoxifen or fulvestrant  needs to be removed from methods, no experiment are shown with those treatment.

Response 6: We thank the reviewer for pointing this out. We have corrected this problem in Methods (Page 12, Line 393-395).

Round 2

Reviewer 2 Report

The authors addressed most of the issues raised and adjusted the text carefully, to not overstate their findings. The manuscript can be published in the current form.   

This manuscript is a resubmission of an earlier submission. The following is a list of the peer review reports and author responses from that submission.

Round 1

Reviewer 1 Report

After the changes made to the manuscript, all major concerns were addressed. This reviewer still disagrees with the authors' view on mammosphere vs. colony formation assays, but this is more a point in style. 

Reviewer 2 Report

The manuscript by Dong et al is a re-submission of a previous manuscript to this journal. The manuscript presents data describing the role of ZBHX3 in breast cancer, ascribing a role of ZBHX3 in transactivation of MYC and TBX3. There are not a lot of new data when compared to the previous version and I still have the following reservations:

I have concern that the data is still over-interpreted in most of the figures. For example, in the western blots of Fig 3f and h, increased MYC expression with increased E2 dose seems to be largely independent of ZFHX3. Western blots are semi-quantitative at best and the only reasonable conclusion that could be drawn from these panels is that, regardless of the level of ZFHX3, MYC expression is increased upon E2 stimulation. The slope of the two curves in Fig 3g is the same and is expressed as % of 0 would be overlapping. This seriously question the conclusions drawn here. There also seems to be variation in the western blots results: compare the bands for ZFHX3 for the cell lines BT474 and MDA-MB-231 in S1 and, respectively, S6 and S5. Compare also the TY47D levels of TBX3 AND MYC in S1 and Fig 3. Those are more than gels-to-gels variations. Further, how can the author conclude that the reduced sphere formation in Fig.5f is due to TBX3 knockdown when no noticeable reduction of expression of TBX3 is seen in Fig 5e?

I am puzzled by the selective use of the cell lines for the silencing and over-expression experiments. MCF-7 is use only for silencing while T-47D solely for over-expression. Are the results due to the gene manipulation or intrinsic to the cell lines. BT-474 would be a good model where both suppression and over-expression of the different genes could be analysed in the same line.

Figure 1i is still in direct contradiction with reference 20 and 24 (the author own paper). This is summarily, and not satisfactorily, dismissed in the discussion.  The authors’ argument about the existence of two splicing variants need to be explored here. The splicing variant A is used in ref. 20 but also here in the ectopic expression here. Variant B could be used in a small number of experiments to explore their relative effect and importance. What variants are targeted by the silencing experiments?

How are the positive cells counted in Fig3 a and b? By just eyeballing them, I am not convinced of the results. TBX3 and Oct4 are at 80 % but the photos look markedly different.

Additional comments:

Figure 1h: assuming this is the T-47D cells, it is surprising that there is so little ki67+ cells in the IHC. The tumour is growing so one would expect some (as is shown in the barchart). Also, the density of the two ZFHX3 versus the ki67 is quite different. Why?

Figure 2c and f: The isotype controls should be shown (in supps) to be convinced that the gates are properly placed.

S3: Clarify the legend to incorporate the meaning of Rep.1, 2… as well as colour code.

Reviewer 3 Report

Thank you to the authors for trying to address the concerns stated in the first review round. Dong et al. describe the potential function of ATBF1/ZFHX3 in stemness via regulation of expression of MYC and TBX3 in ER positive cell lines.

The first major concern was the discrepancy between the earlier published papers and the submitted manuscript. The authors explain the discrepancy in the contradictory results around the function of ZFHX3 based on the splicing variants of ZFHX3 and the composition of the complete medium. It would be advantageous if the shRNA could be designed specifically against each splice variant and the results being included into the manuscript otherwise the discrepancy persists. Additionally, by removing the data from the manuscript which showed the E2 influence in Fig. 1 the main point of E2 regulation and ER-positive cells response is speculative.

The second major concern stated that not the same way of regulation of MYC and TBX3 can be responsible in ER positive cells and TNBC cell line, as the endogenous levels and translocation of ZFHX3 are not the same. As suggested before the authors either should include data of MDA-MB-231 and the translocation of endogenous ZFHX3 to pinpoint the transcriptional regulation of MYC and TBX3 or the corresponding text in results and discussion needs to be removed from the manuscript and TNBC should not be included.

Finally, by addressing the minor concern 5 the authors admit that the effect of stemness and tumorigenicity cannot only be attributed to the activation of MYC and TBX3 by ZFHX3 which the claim in the title of the manuscript. If that is not the case, they should overthink the main message of the manuscript and definitely change the title.

As the authors fail to show that the effect is regulated by the transactivation of MYC and TBX3 the manuscript cannot be published like this. The authors overstate and over interpret the results, as all requested experiments were either dismissed or didn’t show the confirming results. Please consider major revisions before publishing.